# Analysis of miR-29 Serum Levels in Patients with Neuroendocrine Tumors—Results from an Exploratory Study

**DOI:** 10.3390/jcm9092881

**Published:** 2020-09-06

**Authors:** Burcin Özdirik, Anna K. Stueven, Raphael Mohr, Lukas Geisler, Alexander Wree, Jana Knorr, Münevver Demir, Mihael Vucur, Sven H. Loosen, Fabian Benz, Markus Reiss, Bertram Wiedenmann, Frank Tacke, Henning Jann, Teresa Hellberg, Christoph Roderburg

**Affiliations:** 1Department of Hepatology & Gastroenterology, Campus Virchow Klinikum and Campus Charité Mitte, Charité University Medicine Berlin, Augustenburger Platz 1, 13353 Berlin, Germany; burcin.oezdirik@charite.de (B.Ö.); anna-kathrin.stueven@charite.de (A.K.S.); raphael.mohr@charite.de (R.M.); lukas.geisler@charite.de (L.G.); alexander.wree@charite.de (A.W.); jana.knorr@charite.de (J.K.); muenevver.demir@charite.de (M.D.); fabian.benz@charite.de (F.B.); markus.reiss@charite.de (M.R.); bertram.wiedenmann@charite.de (B.W.); frank.tacke@charite.de (F.T.); henning.jann@charite.de (H.J.); teresa.hellberg@charite.de (T.H.); 2Department of Medicine III, University Hospital RWTH Aachen, Pauwelsstrasse 30, 52074 Aachen, Germany; mvucur@ukaachen.de; 3Clinic for Gastroenterology, Hepatology and Infectious Diseases, Medical Faculty, University Hospital, Düsseldorf, Moorenstraße 5, 40225 Düsseldorf, Germany; sloosen@ukaachen.de

**Keywords:** miR-29, neuroendocrine tumor, neuroendocrine carcinoma, biomarker, survival

## Abstract

Background and aims: Due to its involvement in tumor biology as well as tumor-associated stroma cell responses, recent data suggested a potential role of miR-29 as a biomarker for different malignancies. However, its role in neuroendocrine tumors (NETs) is only poorly understood. Methods: We measured circulating levels of miR-29b in 45 patients with NET and compared them to 19 healthy controls. Results were correlated with clinical records. Results: In our cohort of NET patients treated between 2010 and 2019 at our department, miR-29b serum levels were significantly downregulated when compared to healthy control samples. Further, a significant correlation between chromogranin A (CgA) and relative miR-29b levels was noted. However, serum levels of miR-29b were independent of tumor-related factors such as proliferation activity according to Ki-67 index, tumor grading, the TMN stage of malignant tumors, somatostatin receptor expression or clinical features such as functional or non-functional disease and presence of tumor relapse. Finally, in contrast to previous results from other malignancies, miR-29b serum levels were not a significant predictor of overall survival in NET patients. Conclusion: Our data suggest a role for miR-29b serum levels as a previously unrecognized biomarker for diagnosis of NET. However, miR-29 does not allow for predicting tumor stage or patients’ outcome.

## 1. Introduction

In the last decades, intensive research activities have been undertaken to identify biomarkers for diagnosis and for guiding therapeutic decisions in patients with neuroendocrine tumors (NET) [1]. However, so far only Chromogranin A (CgA) levels demonstrated an acceptable specificity and sensitivity to differentiate between patients with NET and healthy individuals [2,3]. However, since prognosis of NET patients is still ambiguous and mainly depends on an early diagnosis, innovative parameters reflecting novel pathophysiological concepts are eagerly awaited to improve the clinical management of patients with NET [4]. The introduction of novel biomarkers into clinical routine is mainly hampered by the rarity of NET. Consecutively, large cohorts allowing a systematic identification of markers in these patients are lacking [5]. Moreover, protein-based markers that have been tested so far in the context of NET bear some important limitations, such as their chemical complexity and their susceptibility to degradation [6].

MicroRNAs (miRNAs) represent a class of small RNAs that do not withhold information to encode for proteins but regulate the expression of their target genes on a posttranscriptional and posttranslational level [7]. By regulating whole networks of genes, miRNAs are involved in manifold physiological and pathophysiological processes including carcinogenesis [8]. Besides their role in the regulation of gene expression, miRNAs have been proposed as diagnostic, prognostic and predictive biomarkers in several human diseases [9]. The regulation of miRNA concentrations specifically in NET is poorly understood, and only a few miRNAs were systemically analyzed on their diagnostic or prognostic value [6]. As an example, a recent pilot study demonstrated that many different serum miRNAs (miR-125b-5p, -362-5p, -425-5p and -500a-5p) are upregulated in small bowel NET and might be used for diagnosis of NET [10].

MicroRNA-29b (miR-29b) is part of the miR-29 family that plays a pivotal role in fibrogenesis and carcinogenesis [7,11,12]. Interestingly, the miR-29 family might act as a tumor suppressor and as a tumor promotor. As a tumor suppressor, e.g., in lung cancer, miR-29 restrains cancer progression by promoting tumor cell apoptosis by suppressing DNA methylation of tumor-suppressor genes, by reducing proliferation of tumors and by increasing chemosensitivity. However, as a tumor promoter, miR-29 mediates epithelial-mesenchymal transition (EMT) and promotes metastasis in breast cancer and colon cancer [11,12,13]. Statements about miR-29b expression in NET in previous literature are contradictive since Yoshimoto et al. report an upregulation in tissue of low-grade gastrointestinal NET and pulmonary carcinoid, while Bowden revealed lower serum levels of miR-29b in tissue of small intestine NET [14,15]. Therefore, in this study we analyzed the diagnostic and prognostic value of miR-29b serum levels in a large cohort of NET patients who were treated at our outpatient unit between 2010 and 2019.

## 2. Materials and Methods

### 2.1. Design of Study and Patient Cohort

In this study, we evaluated circulating levels of miR-29 as a novel diagnostic and/or prognostic biomarker in a cohort of 45 patients with NET that were treated at our institution between 2010 and 2019. MiR-29 concentrations were linked to the patients’ clinical characteristics and outcome. The presence of NET was confirmed histopathologically after biopsy or tumor resection. Patients’ blood samples were collected and were centrifuged for 10 min at 2000 g, and serum aliquots of 1 mL were frozen immediately at −80 °C in order to avoid repetitive freeze–thaw cycles until use. Nineteen healthy blood donors who showed no evidence of a malignant tumor served as control samples. Patients were included into the study upon providing written informed consent. The study protocol was approved by the local ethics committee and conducted in accordance with the ethical standards laid down in the Declaration of Helsinki (ethical approval number EA1/229/17).

### 2.2. miRNA Isolation from Serum

Total RNA was isolated from human serum samples using the miRNeasy Serum/Plasma Advanced Kit (Qiagen, Hilden, Germany) according to the manufacturer’s instruction. After 300 μL serum was transferred into a 2 mL microcentrifuge tube, 90 μL buffer RPL (Qiagen, Hilden, Germany), which contains guanidine thiocyanate as well as detergents that are designed to facilitate lysis and denature protein complexes and RNases, was added, vortexed and incubated at room temperature (RT) for 3 min. To precipitate inhibitors (mostly proteins that are highly concentrated in serum samples), 90 μL buffer RPP (Qiagen, Hilden, Germany) was added and mixed vigorously followed by an incubation period of 3 min at RT. Samples were centrifuged for 3 min at 12,000× *g* (Eppendorf Centrifuge 5415 R, Hamburg, Germany) at RT until complete phase separation. The aqueous phase containing total RNA was precipitated with one volume (350–375 μL) 100% isopropanol. In the next step, the entire sample was transferred to a RNeasy UCP MinElute column and centrifuged for 15 s at 8000× *g*. Then 700 μL buffer RWT (Qiagen, Hilden, Germany) was added, followed by centrifugation for 15 s at 8000× *g* RT. Subsequently, 500 μL 80% ethanol was added, followed by centrifugation at RT for 15-s at 8000× *g*. Precipitated RNA was resuspended in 20 μL RNase-free water. In the next step, RNA was eluted into 20 μL of free water and stored at −80 °C.

### 2.3. Quantitative Real-Time PCR

Quantitative real-time polymerase chain reaction (PCR) was performed as recently described [7,16]. In detail, 5 µL of extracted total RNA was used to synthesize complementary deoxyribonucleic acid (cDNA) utilizing a miScript Reverse Transcriptase Kit (Qiagen, Hilden, Germany) according to the manufacturer’s protocol and was diluted in suitable amounts of H_2_O. The rest of the protocol was conducted via the miScript Reverse Transcription Kit according to manufacturer’s protocol (Qiagen, Hilden, Germany). cDNA samples (2 µL) were used for quantitative real-time PCR in a total volume of 25 µL using the miScript SYBR Green PCR Kit (Qiagen, Hilden, Germany) and miRNA-specific primers (miR-29b, miR-16 (analyzed for data normalization)) on a qPCR machine (Applied Biosystems 7300 Sequence Detection System, Applied Biosystems, Foster City, CA, USA). All results are expressed as 2-ΔΔCT and represent the x-fold increase of gene expression in relation to our housekeeping gene miR-16. Data were generated and analyzed using the SDS 2.3 and RQ manager 1.2 software packages.

### 2.4. Statistical Analysis

Serum data are displayed as scatter plots. We used the Mann-Whitney *U* test or the Kruskal-Wallis test for multiple group comparisons. Correlation analyses were performed using the Spearman’s correlation coefficient. We generated receiver operating characteristic (ROC) curves by plotting the sensitivity (%) against 100% specificity (%). Optimal cut-off values for ROC curves were calculated with the Youden’s Index method (YI = sensitivity + specificity − 1). Kaplan-Meier curves display the impact of a specific parameter on the overall survival. The respective 95% confidence intervals were estimated with the Kaplan–Meier survival method. Survival curves between groups were compared by the Log-rank Mantel-Cox test. All statistical analyses were performed with Prism (version 7.03; GraphPad, La Jolla, California, CA, USA). A *p* value of <0.05 was considered statistically significant (* *p* < 0.05; ** *p*** < 0.01; *** *p* < 0.001).

## 3. Results

### 3.1. Patient Characteristics

Forty-five patients with histologically confirmed NET were included into the present analysis. Out of these, 24 (53%) were female. Median age at initial diagnosis was 59 years (17–80). Primary tumor localizations were ileum (*n* = 23) and pancreas (*n* = 21) as well as the stomach (*n* = 1) in one case. Median time of follow-up was nine years (range 0–21 years) and the median Ki-67 proliferation index was 2% (range 1–50%). Twenty-three (52%) tumors were histologically characterized as Grade 1, 17 (39%) as Grade 2, and 4 (9%) as Grade 3. In total, 18% (*n* = 8) of our patients were untreated and 67% (*n* = 30) received surgery at the time of serum sampling. Only 24% (*n* = 11) received biological treatment with somatostatin analogue (SSA) and/or interferon alpha therapy. One patient received radiotherapy, chemotherapy and peptide receptor radionuclide therapy.

### 3.2. Levels of Circulating miR-29b in NET Patients

Altered serum concentrations of miR-29b were recently demonstrated in various inflammatory, fibrotic and malignant diseases [7,17,18]. We hypothesized that miR-29b levels might also be dysregulated in patients with NET. Of note, we found significantly lower concentrations of circulating miR-29b in the patient group when compared to healthy controls (Figure 1A). We next applied ROC curve analyses to identify the discriminatory power of miR-29b for distinguishing between patients with NET and healthy volunteers. These analyses showed an area under the curve (AUC) of 0.7922 for miR-29b regarding the discrimination between NET and controls (Figure 1B). At the ideal cut-off value of 2.145 (arbitrary units (AU)), miR-29b showed a sensitivity of 93% and a specificity of 56% for the identification of NET. Interestingly, miR-29b levels were further decreased in patients displaying higher levels (above the median value) in chromogranin A when compared to those with lower levels (below the median value) (Figure 1C, Appendix A).

As the role of the miR-29 family in the context of cardiovascular and metabolic diseases was recently demonstrated [19,20], we analyzed whether type 2 diabetes mellitus or arterial hypertension might further alter miR-29b concentrations in patients with NET. However, in our cohort relative miR-29b concentrations were independent of the presence of these metabolic comorbidities (Figure 1D,E). Moreover, miR-29b was independent of the patient’s age or sex (Figure 1F,G).

### 3.3. miR-29b Serum Concentrations Are Not Associated with Disease Characteristics in Patients with NET

Considering the striking difference between NET patients and healthy volunteers, we aimed to find out whether serum concentrations of miR-29b might also reflect disease-specific clinicopathological characteristics. We therefore compared miR-29b levels in patients with different tumor localizations (Figure 2A), with lower or higher Ki-67 rates (Figure 2B), different histological tumor grading (Figure 2C), functional or non-functional disease (Figure 2D), as well as positive or negative somatostatin receptor (SSR) expression status (Figure 2E). Furthermore, we analyzed miR-29b concentrations in patients with advanced or earlier disease (Figure 2F), non-metastatic or metastasized disease (Figure 2G), lymph node positive or negative disease stage (Figure 2H), as well as in patients with/without hepatic (Figure 2I) and bone metastases (Figure 2J). However, all of these different subgroups had similar miR-29b levels, highlighting that circulating miR-29b rather reflects the presence of NET than tumor-specific factors. Finally, we analyzed whether miR-29b concentrations might be associated with the postoperative relapse status (Figure 2K). However, patients with or without tumor relapse showed almost identical relative miR-29b serum levels.

It was recently suggested that circulating miRNAs are cleared by the kidney, and serum concentrations might be altered in patients with impaired kidney function. Nevertheless, in our cohort patients with creatinine concentrations ≥1.5 mg/dL had similar miR-29b levels compared to patients with creatinine concentrations < 1.5 mg/dL (Figure 3A). Moreover, Spearman’s rank analysis did not reveal a correlation between miR-29b and creatinine concentrations in our cohort of patients (Figure 3B).

### 3.4. Circulating miR-29b Levels Do Not Reflect Overall Survival in Patients with NET

Recently published data revealed a prognostic value of circulating miR-29b in different gastrointestinal malignancies [11,12,13]. By applying Kaplan–Meier curve analysis, we aimed to validate the prognostic relevance of circulating miR-29b in our cohort of patients with NET. In this analysis, patients with miR-29b serum levels above/below the median and within the 25th to 75th percentile of all patients showed a similar survival. Moreover, relative miR-29b levels did not correlate with the patient’s survival time, suggesting that miR-29b does not represent a prognostic marker in patients with NET (Figure 4A–C).

## 4. Discussion

We demonstrate that levels of circulating miR-29b are significantly decreased in patients with neuroendocrine neoplasms but are not associated with patients’ clinicopathological characteristics or prognosis. Our data are in line with the results from a previous study of Bowden et al. [15], showing lower levels of miR-29b in sera of a different cohort of patients with (metastatic) neuroendocrine tumors. Interestingly, in our analysis miR-29b levels were further decreased in patients with higher CgA levels but did not reflect the tumor localization, the tumor proliferation activity nor any other clinicopathological tumor features.

These data clearly raise the question of the source of circulating miR-29b in patients with NET. In this context, it is important to note that the specific process between intra- and extracellular miRNA levels are presently unknown. For many miRNAs, contradictory regulations between serum and tissue expression were described, suggesting that levels of circulating miRNAs are rather actively regulated in different diseases [21,22,23,24]. Recent studies provide evidence that miRNAs are packed into exosomes and might be exchanged between the extra- and intracellular miRNA pool [25]. On the other hand, miRNAs might also be passively released during cell death [26].

Recently, members of the miR-29 family were identified as both tumor suppressors and oncogens in very different cancers (summarized in [11,12,13]), including NET [27,28]. MiRNAs regulate gene expression at the post-transcriptional level through a complementary base pairing with the target mRNA, leading to mRNA degradation (in the case of perfect complementation) or translation inhibition (in the case of imperfect complementation) [29]. Based on their tissue-specific expression, their rapid release into the circulation and a remarkable stability in plasma, circulating miRNA are presently scrutinized for their capability as biomarkers for hepatocellular carcinoma both in a diagnostic and prognostic setting [30]. Measurements of circulating miRNAs might serve as a potential new approach for prompt and non-invasive diagnostic/prognostic screening using real-time PCR. Our results from an exploratory analysis of circulating miR-29b establish a previously unrecognized role of this miRNA in patients of NET and should trigger further research in the field. Given the low incidence of NET, biomarker analysis from large and clinically well annotated patient cohorts is scarce [5]. In this context, the data presented here are potentially of high interest, since diagnosis of neuroendocrine neoplasia (NEN) relies on histopathological analysis, in contrast with other diseases where serum-based markers (“liquid biopsy“) are proposed as an easy alternative to histology [4]. 

NEN comprises both NET and neuroendocrine carcinoma (NEC), representing very different tumor entities in terms of pathophysiology, treatment and patient’s prognosis [31,32,33]. Unfortunately, our cohort of patients only consisted of patients with NET and the question of whether NEC is associated with similar changes in miR-29b concentrations cannot be answered here. Both from a clinical and a pathophysiological point of view, such data would be of high interest since, in contrast to NET, chromogranin A is neither diagnostic nor prognostic in NEC. Therefore, novel biomarkers are urgently needed in patients suffering from neuroendocrine carcinoma.

In summary, the data from this exploratory analysis suggest that measurements of circulating miR-29b might be useful as an additional tool in the complex diagnostic work-up of patients with NET. However, our analyses bear some important limitations. First, we applied a single center design and results from this study warrant confirmation in a multi-center approach. Moreover, our study did not include longitudinal measurements during treatment such as chemotherapy or loco-regional therapies and we cannot provide data showing whether the course of circulating miR-29b reflects tumor response or whether a further decrease in miR-29b concentrations might have a different outcome than in patients whose levels increase. Thus, further studies, including larger patient numbers and potentially featuring a multi-center design, need to be performed before the important question of the role of miR-29b in NET can be fully answered.

## 5. Conclusions

Our data suggest a role for miR-29b serum levels as a previously unrecognized biomarker for diagnosis of NET. However, miR-29 does not allow for predicting tumor stage or patients’ outcome.

## Figures and Tables

**Figure 1 jcm-09-02881-f001:**
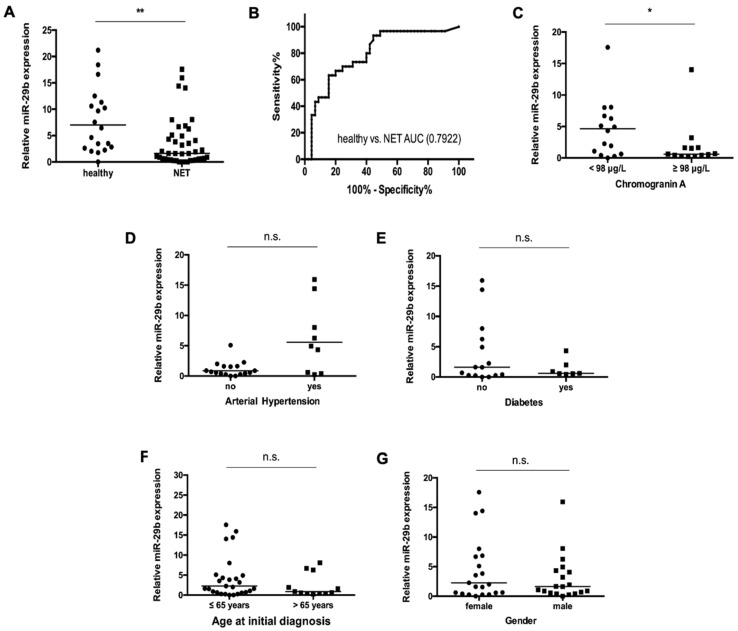
miR-29b levels are decreased in patients with neuroendocrine tumors (NET). (**A**) qPCR was used to determine relative concentrations of circulating miR-29b in patients with NET as well as in healthy controls. (**B**) Relative miR-29b levels display an AUC value of 0.7922 regarding the discrimination of NET patients and healthy controls. (**C**) Relative miR-29b concentrations were lower in patients with higher levels in chromogranin A (above the median value (98 μg/L)) when compared to those with lower levels (below the median value (98 μg/L)). Relative miR-29b concentrations in serum were similar in patients (**D**) with or without arterial hypertension (**E**), with or without type 2 diabetes, and (**F**) patients younger/older than 65 years, and (**G**) did not vary with respect to patients’ gender. The scatter plots display relative miR-29 expression levels between the two subgroups. The black horizontal lines represent the median per group. (* *p* < 0.05; ** *p* < 0.01; *** *p* < 0.001).

**Figure 2 jcm-09-02881-f002:**
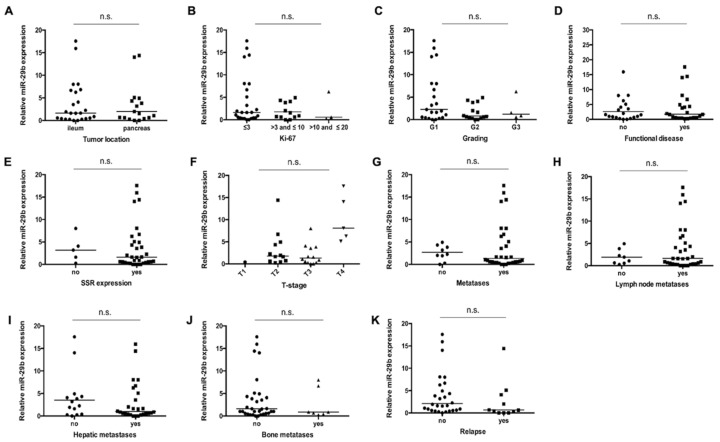
miR-29b expression does not reflect tumor characteristics. Analysis of miR-29 serum levels with respect to (**A**) tumor localization, (**B**) Ki-67 rates, (**C**) histological grading (Grade 1 to 3), (**D**) functional or non-functional disease (**E**) SSR expression, (**F**) T-stages, (**G**) presence of distant metastases, (**H**) presence of lymph node metastases, as well as (**I**) hepatic and **(J**) bone metastases and finally (**K**) postoperative relapse status. The scatter plots display relative miR-29 expression levels between the two subgroups. The black horizontal lines represent the median per group. (* *p* < 0.05; ** *p* < 0.01; *** *p* < 0.001).

**Figure 3 jcm-09-02881-f003:**
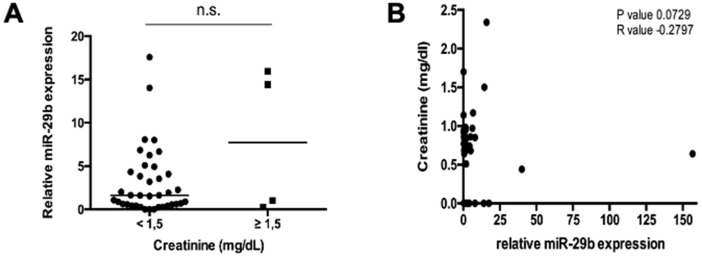
miR-29b levels in NET are not indicative for kidney injury. (**A**) Analysis of miR-29b serum levels with respect to an impaired kidney function. (**B**) Spearman’s rank analysis does not reveal a significant correlation between creatinine values and relative miR-29b concentration in serum levels of NET patients. The scatter plots display relative miR-29b expression levels between the two subgroups. The black horizontal line represents the median per group (* *p* < 0.05; ** *p* < 0.01; *** *p* < 0.001).

**Figure 4 jcm-09-02881-f004:**
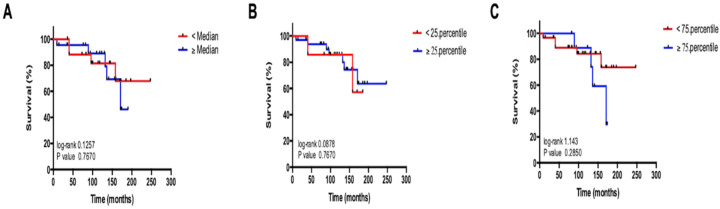
miR-29b expression is not associated with the patients’ prognosis. Kaplan–Meier analysis of serum miR-29b levels above (red curve) and below (blue curve) the (**A**) median (1.64 (AU)), (**B**) the 25th percentile (0.49 (AU)) and (**C**) 75th percentile (6.47 (AU)) of all miR-29b concentrations.

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
