# Peer review of "Analysis of miR-29 Serum Levels in Patients with Neuroendocrine Tumors—Results from an Exploratory Study"

_jcm, 2020, doi:10.3390/jcm9092881_

Round 1

Reviewer 1 Report

In the study by Özdirik et al., the authors explore levels of miR-29b in neuroendocrine tumours. While the study is scientifically sound, it contains, in my opinion, little novelty and its findings would be of little use in clinical practice. The AUC value reported for differentiation between healthy controls and NETs is rather low at this early stage of research (the excellent biomarkers start from the values of 0.9). 

I also have remarks about the text itself:

  1. In the introduction no previous connection between NETs and miR-29b is mentioned - no justification for choice of this miRNA is given except for its previously described role as a biomarker in other cancer types and its role in carcinogenesis. I think authors should expand this section and at least cite a work by Yoshimoto et al. (https://dx.doi.org/10.1159/000461582) who has found upregulation of miR-29b in gastrointestinal NETs. 
  2. In the statistical analysis section the authors mention "Non-parametric data", however the statistics can be non-parametric (i.e. not assuming the data come from particular distribution), not the data. I would suggest adding why the authors think the non-parametric statistics are in order in this case and correcting the statement. 

Author Response

Response letter

We are grateful to the reviewer for his thoughtful and diligent handling of our manuscript. Answering each issue raised, helped us to further improve our manuscript.

1.Response: We thank the reviewer for this important comment. In order to further highlight why miR-29b (and not other miRNAs) were chosen for further analysis, we have included the following sentence into the new introduction section of the revised manuscript:

“Statements about miR-29b expression in NET in previous literature are contradictive since Yoshimoto et. al. report an upregulation in tissue of low grade gastrointestinal NET and pulmonary carcinoid, while Bowden revealed lower serum levels of miR-29b in tissue of small intestine NET (14, 15). Therefore, in this study, we analyzed the diagnostic and prognostic value of miR-29b serum levels in a large cohort of NET-patients who were treated at our outpatient unit between 2010 and 2019.” (page1,l.62-67)

2.Response: We thank the author for this comment.We used nonparametric tests since our data did not fulfill the criteria for parametric tests. Our data are not normally distributed. Moreover, the sample size is quite small and, in many cases, it is smaller than n=20 as it is recommended for performing Mann-Whitney U-test. Since the previous “wording” was not totally correct, we have rewritten the following paragraph:

“Serum data are displayed as scatter plots. Data were compared using the Mann-Whitney U test or the Kruskal-Wallis-Test for multiple group comparisons. Correlation analyses were performed using the Spearman's correlation coefficient.” (page 3, l.108-109)

Reviewer 2 Report

In the design of the study is not repoted the time of blood sample collection (at the timo of the diagnosis? during the treatment?)

Author Response

Response letter

We are grateful to the reviewer for the professional handling of our manuscript.

Response:We want to apologize not to have provided this important information already with the initial version of the manuscript. Wehave now included detailed information on therapies performed until time of serum sampling and during course of disease.

18% (n=8) of our patients were untreated and 67% (n=30) received surgery at time of serum sampling. Only 24% (n=11) received biological treatment with SSA and/or Interferon alpha therapy. Only one patient received radiotherapy, chemotherapy and PRRT.” (page 3, l.125-127)”

Reviewer 3 Report

This study explored Mir29b as a new marker for GEP-NET. This study included a large cohort of patients well characterized. Numerous clinical aspects of the tumors were checked to research a correlation with the miR-29b level. The paper is well written and presented

However I raised 2 points

  1. I don’t understand the figure in comparison to the text : in the text the authors wrote on the increased level of miR-29b in the serum of these patients vs. control population but in all figures the miR-29b levels appeared lower. The authors have to better explained the methods of quantification (page 3 line 10) : “all results are expressed as 2DDct and represent the x fold increase of gene expression compared to the control group” What is the control group ? What is the endogenous gene ?
  2. Line 227 “non invasive diagnostic” : The authors studied the specificity of this marker in patients with neuroendocrine tumors vs. control population without tumors but what about patients bearing NET vs patients bearing other types of pancreatic or ileum tumors ?

Author Response

Response letter

We are grateful to the reviewer for his thoughtful and diligent handling of our manuscript. Answering each issue raised, clearly helped to significantly improve our manuscript. Our answers are as follows:

1.Response:We are grateful to the reviewer for carefully reading the manuscript. Indeed miR-29 is downregulated in patients with NET as shown in all figures. We have amended the manuscript accordingly.

2.Response:In our study we found out that miR-29b levels in serum of NET patients are significantly lower expressed than in serum of healthy volunteers. Our results are expressed as 2-ΔΔCT and represent the x-fold increase of gene expression in relation to our housekeeping gene miR-16.  Since the previous “wording” was not quite clear, we have rewritten this paragraph.

“All results are expressed as 2-ΔΔCT and represent the x-fold increase of gene expression in relation to our housekeeping gene miR-16. Data were generated and analyzed using the SDS 2.3 and RQ manager 1.2 software packages.”(page 3, l.104-106)

3. Response: This is an interesting reviewers’ question. Unfortunately, we do not have access on sera from patients with pancreatic or ileal tumors other than NET since our outpatient unit is dedicated only to the treatment of patients with NET. Therefore, we are unable to perform this analysis.  Nevertheless, we fully agree with the reviewer that such measurements would be of high value.

Round 2

Reviewer 3 Report

The authors have corrected the text according to the figures and better described the methods.